# Safety and Therapeutic Optimization of Lutetium-177 Based Radiopharmaceuticals

**DOI:** 10.3390/pharmaceutics15041240

**Published:** 2023-04-13

**Authors:** Typhanie Ladrière, Julie Faudemer, Elise Levigoureux, Damien Peyronnet, Cédric Desmonts, Jonathan Vigne

**Affiliations:** 1Department of Nuclear Medicine, CHU de Caen Normandie, Normandie Université, UNICAEN, 14000 Caen, France; 2Department of Pharmacy, CHU de Caen Normandie, Normandie Université, UNICAEN, 14000 Caen, France; 3Hospices Civils de Lyon, Groupement Hospitalier Est, 69677 Bron, France; 4Lyon Neuroscience Research Center, CNRS UMR5292, INSERM U1028, Université Claude Bernard Lyon 1, 69677 Bron, France; 5INSERM U1086, ANTICIPE, Normandy University, UNICAEN, 14000 Caen, France; 6PhIND, Centre Cyceron, Institut Blood and Brain @ Caen-Normandie, INSERM U1237, Normandie Université, UNICAEN, 14000 Caen, France

**Keywords:** radiopharmaceuticals, Lutetium-177, nuclear medicine, peptide receptor radionuclide therapy, safety, dosimetry, risk management, nuclear medicine

## Abstract

Peptide receptor radionuclide therapy (PRRT) using Lutetium-177 (^177^Lu) based radiopharmaceuticals has emerged as a therapeutic area in the field of nuclear medicine and oncology, allowing for personalized medicine. Since the first market authorization in 2018 of [¹⁷⁷Lu]Lu-DOTATATE (Lutathera®) targeting somatostatin receptor type 2 in the treatment of gastroenteropancreatic neuroendocrine tumors, intensive research has led to transfer innovative ^177^Lu containing pharmaceuticals to the clinic. Recently, a second market authorization in the field was obtained for [¹⁷⁷Lu]Lu-PSMA-617 (Pluvicto®) in the treatment of prostate cancer. The efficacy of ^177^Lu radiopharmaceuticals are now quite well-reported and data on the safety and management of patients are needed. This review will focus on several clinically tested and reported tailored approaches to enhance the risk–benefit trade-off of radioligand therapy. The aim is to help clinicians and nuclear medicine staff set up safe and optimized procedures using the approved ^177^Lu based radiopharmaceuticals.

## 1. Introduction

Theranostics is an emerging and expanding concept in the biomedical field based on the selection of the patients’ eligibility to therapeutic interventions following a dedicated imaging procedure to verify the presence of a biological target. Iodine-131 was used as the first therapeutic application in nuclear medicine since the 1940s. The availability of Iodine-123 a few years later allowed for the first approach to what would become theranostics based on a pair of radionuclides, one of which is used for diagnosis and the other for therapy [1,2]. Since then, no other radioisotope based theranostic approach (radiotheranostics) has been approved for clinical use until recently [2]. Radiotheranostics is characterized by a combination of a diagnostic tracer targeting a biomarker of interest imaged with positron emission tomography (PET) or single photon emission computed tomography (SPECT). Depending on the target level of expression and localization, the patient can be selected for therapeutic intervention with a corresponding radiopharmaceutical that has the same or similar structure but a different embedded radionuclide. Such a strategy allows for therapy monitoring with an iterative imaging procedure using PET or SPECT [3].

Radiotheranostics represents the most clinically advanced application in the field, with intensive development as highlighted by the considerable investment made by pharmaceutical companies [4,5]. The ideal requirement of radiotheranostics is that the same or similar compound should be able to be labelled with radionuclides dedicated to diagnostic and therapeutic purposes. From a fundamental point of view, it would require two radionuclides of the same chemical element, enabling an accurate theranostic pair that would facilitate the pharmacokinetic assessment to determine the dosimetric estimation of the considered radiopharmaceutical [6,7,8,9]. However, the physico-chemical properties are not the only criteria of choice to select radionuclides to transfer to the clinic, it also depends on parameters such as the possibility of a large scale production, a purification process compatible with human use, and the supply logistics [10,11,12]. 

One of the emerging theranostic radionuclides that best fit with the combination of physico-chemical and production/supply requirements is the radiometal named Lutetium-177 (^177^Lu). First, it presents physical properties suitable for clinical use such as a medium energy β^-^ emission (Emax = 0.5 MeV, 100%) and tissue path length (mean path length = 0.16 mm, maximum path length = 2 mm), which are well-suited to treat disseminated metastatic cancer as it limits the toxicity effects to normal tissue. The therapeutic β^-^ particle of ^177^Lu is emitted together with several accompanying γ-photons of 208 keV (10,4%) and 113 keV (6.2%), which are used for diagnostic and dosimetric purposes. Furthermore, its physical half-life (T_1/2_ = 6.64 days) can be adapted to both supply logistics and clinical use with various targeting agents from peptides to large biomolecules [13,14]. Second, its chemical properties enable chelation with clinically approved bifunctional chelators such as DOTA due to its small ionic radius among lanthanides (0.86–1.03 Å) and a preferred coordination number between 8 and 9 [11]. Third, ^177^Lu decays to form the hafnium-177 stable daughter, and it can both be obtained via cyclotron and nuclear reactor production routes, facilitating its widespread use [12,15].

The first application of ^177^Lu labelled radiopharmaceuticals that have reached the clinics is based on peptide ligands referring to a theranostic approach called peptide receptor radionuclide therapy (PRRT). PRRT is a targeted therapeutic strategy using radiolabeled peptides as targeting vectors designed to deliver cytotoxic levels of radiation dose at the level of the corresponding receptor overexpressed by cancer cells [16,17]. After two decades of clinical investigations in PRRT, it still raises interest because of its ability to treat with high sensitivity and specificity at the molecular level. The first clinical application of ^177^Lu based PRRT was designed to target cancers that overexpressed somatostatin receptors type 2 (SSTR2) [1]. This target is of major interest in neuroendocrine tumors where it is largely overexpressed by cancer cells. The radiotheranostic approach targeting SSTR2 is based on the therapeutic radiopharmaceutical [¹⁷⁷Lu]Lu-DOTATATE (Figure 1a) and its diagnostic counterpart embedding the positron emitter Gallium-68 (^68^Ga) named [^68^Ga]Ga-DOTATATE or [^68^Ga]Ga-DOTATOC. The radiotheranostic pair ^177^Lu/^68^Ga is currently a well-established couple of radionuclides with widespread use in clinical radioligand therapy. In 2017, Strosberg et al. published the results of a randomized controlled clinical trial aiming to evaluate the efficacy and safety of [¹⁷⁷Lu]Lu-DOTATATE in patients with advanced midgut neuroendocrine tumors. From an efficacy point of view, they showed that patients treated with [¹⁷⁷Lu]Lu-DOTATATE presented a longer progression free-survival and a significantly higher response rate than patients treated with the best supportive care alone. From a safety point of view, clinically significant myelosuppression occurred in fewer than 10% of patients treated with [¹⁷⁷Lu]Lu-DOTATATE [18]. In 2018, [¹⁷⁷Lu]Lu-DOTATATE received approval from both the U.S. Food and Drug Administration (FDA) and the European Medicines Agency (EMA) under the commercial name Lutathera® (Advanced Accelerator Applications, a Novartis company). 

Another well-advanced radiotheranostic application relies on the targeting of prostate specific membrane antigen (PSMA) in the context of prostate cancer. PSMA is a type II membrane protein exhibiting enzymatic activity as a glutamate preferring carboxypeptidase [19,20] that is expressed in 85–95% of patients with late-stage prostate cancer. A urea-binding motif small molecule containing a DOTA bifunctional chelator and ^177^Lu called [¹⁷⁷Lu]Lu-PSMA-617 (Figure 1b) was proposed in metastatic castration-resistant prostate cancer (mCRPC) and progressive disease patients after standard treatments. Corresponding results published by Hofman et al. in 2018 suggested that patients treated with [¹⁷⁷Lu]Lu-PSMA-617 had high response rates, as evidenced by a 50% minimum decrease in the prostate specific antigen (PSA) biomarker [21]. In 2021, Sartor et al. reported the results of an international open-label phase 3 trial evaluating [¹⁷⁷Lu]Lu-PSMA-617 in mCRPC patients treated with at least one androgen–receptor-pathway inhibitor and one or two taxane regimens. Importantly, all patients were screened for eligibility using a diagnostic companion drug called [^68^Ga]Ga-PSMA-11 to verify PSMA positivity using PET. The authors concluded that radioligand therapy with [¹⁷⁷Lu]Lu-PSMA-617 prolonged imaging-based progression-free survival and overall survival when added to the standard care of patients with advanced PSMA-positive mCRPC. Additionally, the incidence of adverse events of Grade 3 or above was higher with [¹⁷⁷Lu]Lu-PSMA-617 than without (52.7% vs. 38.0%), but there was no statistical difference regarding quality of life [22]. In the last months, [¹⁷⁷Lu]Lu-PSMA-617 was approved by the FDA and the EMA under the name Pluvicto® (Advanced Accelerator Applications, a Novartis company). Recently, a randomized, parallel group, open-label phase 2 study concluded that [¹⁷⁷Lu]Lu-PSMA-617 is safe and non-inferior to docetaxel in treatment of mCRPC patients. These findings suggest that [¹⁷⁷Lu]Lu-PSMA-617 could be used earlier in the disease course and may then be prescribed to an extended number of potential patients [23]. Therefore, an increasing demand for Lutetium-177 based radiopharmaceuticals should be anticipated, which would require specific training for the nuclear medicine staff.

The present review will focus on several clinically tested and reported tailored approaches to enhance the risk–benefit trade-off of radioligand therapy. The aim is to help clinicians and nuclear medicine staff set up safe and optimized procedures using the approved Lutetium-177 based radiopharmaceuticals.

## 2. Implementation of Lutetium-177 Therapy

### 2.1. Patient Eligibility

#### 2.1.1. [^177^Lu]Lu-DOTATATE (Lutathera®)

Patients with advanced midgut neuroendocrine tumors who have had disease progression during first-line somatostatin analog therapy have limited therapeutic options [24,25,26]. With the exception of everolimus for the treatment of nonfunctional neuroendocrine tumors, no standard second-line systemic treatment options are currently available [27,28]. NETTER-1, a phase 3 trial that was carried out at 41 centers in eight countries across Europe and the USA, allowed for the evaluation of the place of Lutathera® in neuroendocrine tumors in patients aged 18 years and older with advanced, inoperable, well-differentiated (Ki67 index ≤20%) midgut NETs with positive uptake on ¹¹¹In -DTPA octreotide scintigraphy (OctreoScan®) or on [^68^Ga]Ga-DOTATATE/DOTATOC PET-CT on all target lesions and centrally confirmed disease progression on CT or MRI (as per the Response Evaluation Criteria in Solid Tumors [RECIST], version 1.1) while taking a fixed dose of long-acting octreotide 20−30 mg every 3–4 weeks for at least 12 weeks before randomization. Patients had to have a Karnofsky performance status score of at least 60. Previous PRRT was not allowed, nor was any surgery, transarterial therapy, or chemotherapy within 12 weeks of randomization [18,29]. The indication retained in the marketing authorization is the treatment of inoperable, metastatic, progressive, well-differentiated (G1 and G2) gastroenteropancreatic neuroendocrine tumors (GEP-NETs) expressing somatostatin receptors [30]. Lutathera® is a second-line treatment after disease progression with octreotide in metastatic, progressive, well-differentiated, somatostatin receptor-expressing intestinal NETs or a third-line treatment after everolimus failure [31,32,33,34,35,36,37,38]. Eligible patients must have had positive imaging with [^68^Ga]Ga -DOTATATE or [^68^Ga]Ga-DOTATOC to confirm somatostatin receptor overexpression. Patient eligibility should also take into account other treatment options as well as factors that may affect the response to treatment (tumor burden, primary tumor site or previous treatments) [39,40,41,42,43,44,45]. The guidelines published since 2013 [35,37] retain the need for a multidisciplinary meeting (oncologists, nuclear physicians, radiologists, radiopharmacists, surgeons, gastroenterologists, and anatomopathologists) to validate the patient’s eligibility for treatment with [¹⁷⁷Lu]Lu-DOTATATE [40,44,46,47]. Contraindications to treatment are pregnancy, hypersensitivity to the active substance or to one of the excipients, severe associated disease (severe cardiac, hepatic, or hematological disorders), and severe psychiatric disorders. [^177^Lu]Lu-DOTATATE is contraindicated in patients with severe renal impairment and creatinine clearance <30 mL/min. It is not recommended in patients with a creatinine clearance at initiation <40 mL/min [30,37,40]. The ENETS Consensus Guidelines for the Standards of Care in Neuroendocrine Neoplasms recommend that patients have a creatinine clearance ≥50mL/min [35,40].

#### 2.1.2. [^177^Lu]Lu-vipivotide Tetraxetan (Pluvicto^®^)

Pluvicto® is indicated for the treatment of adult patients with progressive metastatic castration-resistant prostate cancer (mCRPC) who have been treated with androgen channel blocking hormone therapy and taxane-based chemotherapy [48].

Pluvicto® was studied as a targeted radioligand therapy (RLT) in an international phase 3 trial named VISION. The VISION trial was a randomized, multicenter, active-control study comparing the best standard of care with or without Pluvicto® [22]. 

The study enrolled men with mCRPC that were in progression after one or more approved androgen–receptor-pathway inhibitor therapies and with either one or two taxane regimens. Eligible patients had PSMA-positive metastatic castration-resistant prostate cancer, in other words, at least one PSMA-positive metastatic lesion (PSMA-positive status was determined with the use of Gallium-68 (^68^Ga)–labeled PSMA-11 ([^68^Ga]Ga-PSMA-11) PET–CT imaging at baseline) [49]. The presence of PSMA-positive lesions was defined in the protocol as a [^68^Ga]Ga-PSMA-11 uptake greater than that of liver parenchyma in one or more metastatic lesions of any size in any organ system. A performance-status score of 0 through 2 (on a scale from 0 to 5, with higher numbers indicating greater disability) [50], a life expectancy of at least 6 months, and adequate organ and bone marrow function were also required [51,52,53,54,55]. 

### 2.2. Procedure of Treatment

#### 2.2.1. [¹⁷⁷Lu]Lu-DOTATATE (Lutathera®)

On the day of injection, the patient is placed in a radiation-protected room sealed by the medical staff (nurse, nursing assistant, or nuclear medicine technician). The floor, especially in the bathroom, should be chosen to facilitate cleaning in the case of vomiting or splashes. It is also possible to cover the floor with absorbent sheets in the case of vomiting. Zoning of the room (i.e., a delimitation and signposting of the restricted zones, allowing the danger of exposure to ionizing radiation to be identified) must be carried out by a competent person in charge of radiation protection (Figure 2) [56]. Patients do not need to fast before treatment.

A fixed dose of 7.4 GBq (200 mCi) of [¹⁷⁷Lu]Lu-DOTATATE is infused intravenously over a period of 30 min. Patients receive four infusions approximately every 8 weeks (cumulative radioactivity, 29.6 GBq [800 mCi]). Reassessment of the clinical and biological status should be routinely undertaken between treatments [18,40,57]. Some patients may continue to receive supportive care with long-acting octreotide, administered intramuscularly at a dose of 30 mg approximately 24 h after each infusion of [¹⁷⁷Lu]Lu-DOTATATE, and then monthly after the completion of four courses [18,40,57] (Figure 3). A post-therapy whole body scan can be performed 24 h after the injection to check the tumor targeting and to obtain an initial idea of the therapeutic response [57].

For patients with clinical symptoms (diarrhea and flushing) and/or adverse events (defined as Grade 2 platelet count toxicity, Grades 3 or 4 hematological toxicity other than lymphocytopenia, a 40% increase in serum creatinine from the baseline with a concomitant decrease of more than 40% in creatinine clearance, or any other Grades 3 or 4 toxicity that may be related to the study drug), infusions may be extended for up to 16 weeks to allow for the resolution of acute toxicity. After the resolution of adverse events, 50% of the standard treatment dose (i.e., 3.7 GBq–100 mCi) can be prescribed [18,30] (Table 1).

Short-acting octreotide is not allowed for 24 h before the administration of [^177^Lu]Lu-DOTATATE, unless this is clinically impossible. Short-acting octreotide can only be continued if the tumor uptake on SSTR imaging during continued somatostatin analog medication is superior to the liver uptake.

#### 2.2.2. [^177^Lu]Lu-Vipivotide Tetraxetan (Pluvicto^®^)

As with Lutathera®, the patient is placed, in the morning, in a radiation protected room or dedicated room with a toilet connected to the radiation decay tanks by medical staff (nurse, nursing assistant or nuclear medicine technician) (Figure 2). Patients do not need to fast before treatment.

The patient receives intravenous infusions of [^177^Lu]Lu-PSMA-617 at a dose of 7.4 GBq (200 mCi) once every 6 weeks (± 1 week) up to a maximum of six cycles, unless there is disease progression or unacceptable toxicity [22,48,51,52,58]. In the case of a patient with a serious adverse event, Pluvicto® may be withheld until improvement or the dose may be reduced by 20% to 5.9 GBq (160 mCi). If these effects persist, treatment may be discontinued (Table 2) [48,55,58]. 

Before any injection of Pluvicto® and between doses (approximately every three weeks), a biological and clinical check-up should be performed [48,51,55,58]. In addition, in non-surgically castrated patients, chemical castration with a gonadotropin-releasing hormone analog should be maintained [48] Patients do not need to fast before treatment.

Monitoring of the PSA biomarker should be carried out throughout the treatment period. A post-injection scan should be performed to confirm the radiopharmaceutical uptake and therefore the tumor uptake [51,55].

### 2.3. Premedication and Administration

#### 2.3.1. [¹⁷⁷Lu]Lu-DOTATATE (Lutathera®)

For renal protection, the most commonly used intravenous amino acid solution is one containing 25 g arginine and 25 g lysine, diluted in 1 l of 0.9% NaCl. This infusion is started about 30 min before the Lutathera® infusion and continues for 4 h [45,57,60,61] (Figure 3). Other infusion protocols exist as well as other commercial solutions of amino acids, but are often more osmolar and therefore more at risk of adverse effects such as nausea and vomiting [45,61,62].

To prevent the onset of nausea or vomiting, premedication with antiemetic drugs is administered 30 min before the infusion of the amino acid solution by an intravenous bolus of granisetron (3 mg), ondansetron (8 mg and most commonly used), or tropisetron (5 mg) [45,57,61]. 

The administration of Lutathera and its associated pre-medication requires collaboration between different health care professionals, (e.g., nurses, nursing assistants, and nuclear medicine technicians).

Different methods of administration are reported and the three main ones are presented herein (Figure 4) [61]. Variations exist but we will not dwell on them here. For gravity administration, during the infusion, the flow of the 9 mg/mL (0.9%) sodium chloride infusion solution increases the pressure in the Lutathera® vial, which facilitates the flow of Lutathera® into the catheter inserted in the patient’s peripheral vein. The amino acid solution is passed through another catheter in the patient’s other arm. For the pump and vial method, the pump is set to deliver the volume of [¹⁷⁷Lu]Lu-DOTATATE over 25–30 min. The [¹⁷⁷Lu]Lu-DOTATATE is delivered in 20–25 mL of volume, resulting in an administration rate of 0.8–0.9 mL/min. Then, there is the pump method with a syringe that will be inserted into a syringe guard. At the end of the infusion, the therapy bottle is rinsed with sterile saline solution [30,45,61,63,64]

#### 2.3.2. [^177^Lu]Lu-Vipivotide Tetraxetan (Pluvicto^®^)

Prophylactic treatments may include antiemetics (e.g., ondansetron) and corticosteroids (such as dexamethasone). Optional treatments include allopurinol in patients with gout or high tumor burden, cooling of the salivary glands before and/or during administration, and the infusion of 500 mL of 0.9% saline after administration [48,51,55,59].

[^177^Lu]Lu-PSMA-617 is administered slowly by the intravenous route. The infusion is preceded and followed by a saline flush with ≥10 mL of sterile 0.9% sodium chloride administered to ensure patency of the IV line and to minimize the risk of extravasation. Vital signs are monitored 15 min before administration and 30 and 60 min after administration. Patients are encouraged to maintain adequate fluid intake to reduce radiation exposure to the bladder [48,51,55]. 

Pluvicto® is administered intravenously by syringe with a syringe shield (with or without a syringe pump), by gravity infusion (with or without an infusion pump), or by vial infusion (peristaltic infusion pump) (Figure 4). Pluvicto® should not be injected directly into another intravenous solution. If a reduced dose of [^177^Lu]Lu-PSMA-617 is available, it should be administered by the syringe or vial method with a peristaltic pump. The gravity method is not recommended as the volume administered may be incorrect [48,59].

## 3. Radioprotection and Risk Management 

Lutetium-177, used in peptide receptor radionuclide therapy (with Lutathera® or Pluvicto®), presents risks of volatility, external and internal exposure, and contamination not only for the various health professionals who handle it, but also for patients and their entourage as well as the public. A series of measures must therefore be taken in terms of radiation protection, prevention, and the management of the risks associated with exposure to this ionizing radiation. 

In 2020, the French Nuclear Safety Authority (ASN) published new recommendations for ^177^Lu [66]. A minimum hospitalization of 24 h is no longer required, and patients can now be admitted to a room, box, or radioprotected area for at least 6 h after the end of the administration. This day hospitalization is only possible when the patient’s clinical condition allows it. During this period, the patient will be required to urinate frequently in a toilet connected to a decay tank that will remove much of the activity and reduce the dose rate to allow discharge. If the patient’s condition does not allow for this day hospitalization, they should be hospitalized for at least 24 h [59,66]. In addition, the service must be able to cope with delays and also make arrangements to receive the patient at night if necessary [67]. 

Whether with Lutathera® or Pluvicto®, the exposure to ionizing radiation of the public, nursing staff, or members of their entourage must be taken into account. For the exposed public, the radiation exposure must be below a limit of 1 mSv/year. The patient and the members of their entourage must be fully informed and warned of the risks concerning the treatment. The ASN has validated a dose constraint for persons involved in patient support, proposed by the GPMED (Permanent Advisory Group on Radiation Protection for Medical and Forensic Applications of Ionizing Radiation), of 3 mSv over 12 months. With Lutathera® and Pluvicto®, since patients receive several treatment cycles per year, the exposure of the patient’s relatives (family, colleagues, etc.) must be kept below one sixth of the annual dose limit. For children and pregnant women, the dose limit remains 1 mSv/year. Nursing mothers should not be forgotten and should also be informed of the risks. Another dose limit has been proposed for health professionals, namely, 5 mSv/year [66,67,68]. To minimize the risk of contamination of the nursing staff as well as the risk of external exposure, various protective measures exist such as the use of vial protectors, syringe protectors, waste and storage containers as well as the use of gloves, tongs, and protective trays [67,68].

After the administration of a ^177^Lu radiopharmaceutical, a high level of radiation is found in the urine up to 48 h after administration. The high excretion rate of [¹⁷⁷Lu]Lu-DOTATATE or [¹⁷⁷Lu]Lu-PSMA-617 should be taken into account to minimize the exposure to ionizing radiation. After 4 h, approximately 50% of the activity is excreted by the kidneys and after 12 h, approximately 70% of the activity is excreted [67,69]. The patient and his family must therefore follow certain rules. These will be applied as soon as the treatment is administered to the patient and will evolve over time (Table 3) [66,68,70]. Radiation protection rules are also necessary for urinary incontinent patients. During the 2 days following the treatment, special precautions must be taken to avoid any radioactive contamination, particularly when handling any material potentially contaminated by urine [30,59].

Finally, with regard to radiation protection, the use of personal or public transport is not recommended for more than one hour during the first three days following treatment [63]. Waste is stored for degradation and disposal in a specific locked and ventilated room [67]. The effluents collected in the degradation tanks present in the department can be eliminated after 10 periods of ^177^Lu[66,67,68]. It is important to recall that ^177^Lu can be produced directly (^176^Lu (n,γ) reaction) without long-lived contaminants, or indirectly as a decay product of the neutron irradiation of ytterbium-176 (^176^Yb (n,γ) reaction). The latter produces ^177^Yb that decays to ^177^Lu and small amounts of metastable Lutetium-177 (^177m^Lu) with a half-life of 161 days, which may require specific regulatory attention depending on the policies of the local authorities [71]. This is the production method used for [^177^Lu]Lu-DOTATATE, which requires more specific waste management. The amount of ^177m^Lu contained in the vial is provided by the supplier. Finally, for deceased patients, appropriate measures for the care of the person are required (decontamination, investigation into the occurrence of death, etc.) as well as the intervention of a competent person in radiation protection [67].

In all cases, the objective is to respect the ALARA (as low as reasonably achievable) principle, a general protection principle that aims to keep exposures to ionizing radiation as low as reasonably achievable.

In addition, the use of ^177^Lu requires the implementation of a quality management system aimed at preventing and managing the risks associated with ^177^Lu exposure. This system must implement the principles of justification and optimization to always be in accordance with the ALARA principle. It must not only include quality assurance in radiation protection [72], but also safe procedures that must be available for all unsealed radiopharmaceutical sources and that cover the required instrumentation, calibration and frequency, ordering and receipt of these products, the content of patient records, and the use and disposal of the wastes of these therapeutic radiopharmaceutical products [63].

A priori risk mapping is also required for all stages of the patient circuit [72]. It must also cover the management of incidents that may occur during the administration of various radiopharmaceutical drugs (Figure 5) [66]. This risk analysis must also take into account the use of various medical devices such as electric syringe pumps. Safety barriers, whether material, human, or organizational, must be put in place to reduce these risks, and in particular, their consequences [72].

Procedures concerning external services or the conduct to be adopted in the face of possible changes (medical devices, premises, treatment practices, etc.) as well as patient monitoring methods must be part of this quality management system [72].

The training of the various health professionals included in this quality management system (initial or continuous) is mandatory and must include recommendations on good radiation protection practices, the preparation of radiopharmaceuticals, and the use of a new medical device as well as the various measures to be implemented to optimize their radiation protection [45,63,66,68,72].

Quality assurance also includes specific procedures for the management of special cases such as patients on hemodialysis, with urinary catheters, or those requiring special assistance [30,48,63,68]. Similarly, it also includes the procedures for recording and analyzing events that may lead to the accidental or non-accidental exposure of the patients to ionizing radiation [63,68,72]. 

This quality management system is therefore at the heart of the therapeutic management of the patient and therefore requires the coordination of all health professionals involved in the patient circuit (Figure 6) [38,72].

One of the most important risks that needs to be addressed is the risk of extravasation. In fact, clinical management of these extravasations requires the preliminary estimation of the dose distribution in the extravasation area. Perfusion must be immediately interrupted, the delivery device must be removed, and the residual activity must be evaluated. The nuclear physician, medical physicist, and radiopharmacist must all be informed.

The area of extravasation should be marked with an indelible pen and a photograph taken, if possible. It is also recommended to record the extravasation time and an estimate of the extravasated volume [73].

The strategy adopted must allow for the simultaneous estimation of the absorbed dose and local treatment of extravasation. The absorbed dose can classically be determined by SPECT/CT imaging immediately after extravasation; SPECT/CT scans can be conducted at day 5 and day 7 for the follow-up. Other methods have been proposed for the estimation dose: Mazzara et al. [74] proposed an analytical model of Lutathera® infusion and extravasation based on EDR (equivalent dose rate) monitoring and Tilsky et al. [75] presented an example of a methodology to estimate the dose distribution after an extravasation.

In order to eliminate dispersion of the product and to prevent stagnation in the tissues, it is recommended to increase the blood flow by elevating the affected arm and applying gentle massage. Depending on the case, aspiration of the extravasated fluid or the application of hot compresses or heating pads to the infusion site to accelerate vasodilation may be considered. Management of ^177^Lu extravasation may include being referred to a plastic surgeon to treat cutaneous lesions [51,76].

It is not only radiopharmaceuticals that can cause extravasation. Extravasation may occur with the amino acid solution used for renal protection during the administration of [^177^Lu]Lu-DOTATATE. Due to its hyper-osmolarity, it can cause serious skin lesions that must be managed by a plastic surgeon [77,78].

In all cases, the patient must be informed. 

## 4. Safety and Pharmacovigilance

The risk–benefit trade-off of the two approved ^177^Lu-labeled radiopharmaceuticals can be considered favorably worldwide. Acute toxicity of these treatments should consider not only the radiopharmaceutical tolerance, but also the safety related to the associated drugs such as the co-infusion of amino acids in the case of [¹⁷⁷Lu]Lu-DOTATATE PRRT. Acute averse events associated with [¹⁷⁷Lu]Lu-DOTATATE and their management are reported in Table 1, but it is important to highlight that amino acid coadministration is susceptible to inducing metabolic acidosis linked adverse events such as nausea, headache, and rarely, vomiting in a majority of patients [79]. Therefore, a check of the ionogram should be performed to prevent electrolyte imbalance and the management of the accompanying side effects of metabolic acidosis should be treated by hydrating the patient with saline and possibly additional corticosteroid and antiemetic administrations [45]. Additionally, the medical staff should be aware of possible hormonal release-induced crises also called carcinoid crisis, which is a medical emergency caused by the excessive release of metabolically active amines than can occur spontaneously after [¹⁷⁷Lu]Lu-DOTATATE administration. The latter adverse event is most likely to occur after the first administration and is reported to be infrequent; its prevention is based on vital signs monitoring (at least blood pressure and pulse) especially for symptomatic patients, and its management relies on high-dose somatostatin analogs, i.v., i.v. fluids, and corticosteroids [80].

Delayed side effects may also occur following PRRT and long-term safety studies have been conducted to better characterize the complete safety profile of [¹⁷⁷Lu]Lu-DOTATATE [29,81]. Renal toxicity is of major importance for the long-term safety profile as kidneys are the dose limiting organs in [¹⁷⁷Lu]Lu-DOTATATE PRRT. It was shown that patients presented a mean creatinine clearance loss of about 3.8% per year despite proper kidney protection [82]. Different parameters such as cumulative renal radiation dose, per-cycle renal radiation dose, age, hypertension, and diabetes are likely to contribute to the decline in renal function following PRRT. One serious but infrequent complication of [¹⁷⁷Lu]Lu-DOTATATE therapy is the occurrence of myeloid neoplasms including acute myeloid leukemia (AML) and myelodysplastic syndrome (MDS). Goncalves et al. reported from a single institution series of 521 PRRT treated patients that the median overall survival was 62 months, but only 13 months for patients that presented therapy-related myeloid neoplasms with death attributed primarily to hematological disease progression [83]. Data from the World Health Organization pharmacovigilance database indicate that AML and MDS represent around 1% of declared side effects with [¹⁷⁷Lu]Lu-DOTATATE, and are significantly higher reported compared to other drugs [84]. There is a need to identify pre-existing genetic factors and predictive biomarkers to minimize the risk of PRRT related myeloid neoplasm. 

Regarding [¹⁷⁷Lu]Lu-PSMA-617, different studies have identified the commonly encountered adverse effects that have been proven to be linked to radiation exposure to non-target tissues, especially salivary glands, lacrimal glands, and kidneys [85,86,87]. The most frequent observed toxicity was Grade 1/2 xerostomia [22]. Indeed, among the salivary glands, a major uptake of [¹⁷⁷Lu]Lu-PSMA-617 is detected in parotid and submandibular glands [88]. Unless salivary glands exhibit PSMA expression, it does not fit with the high uptake observed with diagnostic PSMA-targeted PET/CT scans [89]. Recently, Lucaroni et al. suggested that the unwanted accumulation of [¹⁷⁷Lu]Lu-PSMA-617 in salivary and also in the kidneys may be due to the membranous expression of glutamate carboxypeptidase III (GCPIII), which shares a very high similarity with PSMA in humans [90]. The kidney GCPIII expression, the physiological expression of PSMA in proximal tubular cells [91], and the predominant renal excretion of [¹⁷⁷Lu]Lu-PSMA-617 have raised concerns regarding the possible acute and delayed radiation toxicity to the kidneys. However, there are no consistent data affirming acute or delayed renal toxicity following [¹⁷⁷Lu]Lu-PSMA-617 RLT, even in patients with compromised baseline kidney function [92] or with a single functioning kidney [93]. It remains to be assessed whether potential risk factors such as the number of therapeutic cycles, prior obstructive uropathy, and the loss of renal cortical mass due to the age of the treated population may favor the risk of progressive renal impairment to fully characterize the renal safety profile of [¹⁷⁷Lu]Lu-PSMA-617. The VISION study, a phase III study, also showed hematological toxicity with [^177^Lu]Lu-PSMA with the frequent occurrence of anemia, thrombocytopenia, leucopenia, or lymphopenia [22,48]. High bone tumor burden, previous treatment with taxane-based chemotherapy or pre-existing Grade 2 cytopenia have been identified as risk factors for the development of myelosuppression [94].

## 5. Internal radiotherapy with ^177^Lu: Toward Personalized Dosimetry

The objective of internal vectorized radiotherapy is to deliver the maximum dose to the tumor while sparing healthy tissues. At present, treatments using radiopharmaceuticals labeled with Lutetium-177, whether for the treatment of neuroendocrine tumors or prostate tumors, consist of the administration of fixed doses over a defined number of cycles. The dosage is set empirically and is therefore not adjusted specifically for each patient. Nevertheless, there are strong variabilities between patients following the administration of a radiopharmaceutical related to the rate of tracer uptake by the tumor and the organs at risk. Thus, a review of the literature shows that the minimum and maximum absorbed doses assessed to tumors and healthy tissues in different patients can vary by a factor of 3 and even up to 9 within the same study [95]. The goal of personalized dosimetry is to adapt the injected dose to the patient by evaluating the absorbed doses to the tumor and healthy tissues. This technique allows the treatment to be optimized in terms of both its efficacy and safety.

Although ^177^Lu therapies specifically target receptors overexpressed by the tumors to be treated (i.e., somatostatin receptors for neuroendocrine tumors and PSMA receptors for prostate tumors), healthy organs may also physiologically express this type of receptor and thus generate radiotoxicity related to the accumulation of the tracer in these tissues. The dosimetric evaluation generally focuses on the organs at risk, which are the kidneys, the liver, and the large salivary glands. In addition, since the product is injected by i.v, the blood compartment as well as the bone marrow can be significantly exposed to radiation.

Many authors have already implemented individualized dosimetry for ^177^Lu targeted radionuclide therapy [96,97,98,99,100,101,102,103]. The methodology generally used to calculate the absorbed dose delivered to the organs is that proposed by the MIRD committee [104,105,106]. This formulation is based on the fact that the absorbed dose delivered to a target organ *r_t_* results from the sum of the radiation exposure delivered by a set of source organs *r_s_* that have accumulated radioactivity over time. An organ can, of course, be both the source and target. In order to evaluate the absorbed dose at the tissue level, it is thus necessary to quantify the activity of the tracer present in the different source organs at different times after injection. This allows for a time activity curve (TAC) to be generated whose area under the curve represents the cumulative activity *Ã_S_* in the different source organs, as shown in Figure 7a. The residence time *τ_S_* for each source organ can then be calculated by normalizing the cumulated activity by the total activity *A*_0_ administered to the patient, as follows: $$\tau_{s}=\frac{{\tilde{A}}_{s}}{A_{0}}$$

The absorbed dose to a target organ, resulting from the sum of the radiation exposures delivered by a set of source organs is then calculated as follows:$$D_{r_{s}}=A_{0}\underset{r_{s}}{\sum }\tau_{s}\;S\left(r_{t}\leftarrow r_{s}\right)$$

With $S\left(r_{t}\leftarrow r_{s}\right),$ the *S* factor corresponding to the mean absorbed dose per cumulated activity is defined for different pairs of target and source organs. These calculated values were tabulated for different radiopharmaceuticals and different anthropomorphic phantoms representing different ages and gender models.

Activity measured at the organ level of each organ was carried out using planar or SPECT imaging performed at different times after administration of the radiopharmaceutical. The protocols currently used generally recommend at least three points of measurement, typically one point on the day of treatment, one on the day after, and a late point typically around 7 days. The volume of interest (VOI) was drawn on all organs of interest to measure the mean count per pixel recorded at each acquisition time. A prior calibration of the camera allowed us to convert the number of count per pixel into a radioactive concentration in Bq/mL. The dosimetric workflow is schematized in Figure 7b. All dose calculations can nowadays be performed using commercially available software. Some authors have been able to test these different dosimetry methods and software solutions by analyzing the variability of the calculated doses delivered to patients [107,108,109].

Dosimetry appears to be an essential solution for the management of patients in theranostics. It is already the subject of recommendations from international nuclear medicine societies [95]. However, this approach requires a number of prerequisites to be put in place. First, clinical dosimetry is time consuming, particularly during the implementation of the practice and the training of the different actors involved in the patient’s care. It requires a multidisciplinary approach with the involvement of a medical physicist expert to manage the dosimetry part. Finally, it implies that the patient undergo additional imaging examination at different times after the first treatment cycles.

## 6. Conclusions

Lutetium-177 based radiopharmaceuticals have reshaped the personalized medicine toolbox in nuclear medicine and oncology, but its successful clinical implementation requires a multidisciplinary team approach with appropriate training on the different aspects of radiopharmaceuticals and patient management. The clinical use of radiotheranostics can be more complex than the use of conventional chemotherapy because of the logistical challenges and regulatory hurdles. The expected field expansion requires a global approach and well-defined roles within the implicated health care professionals to set up safe and optimized procedures.

## Figures and Tables

**Figure 1 pharmaceutics-15-01240-f001:**
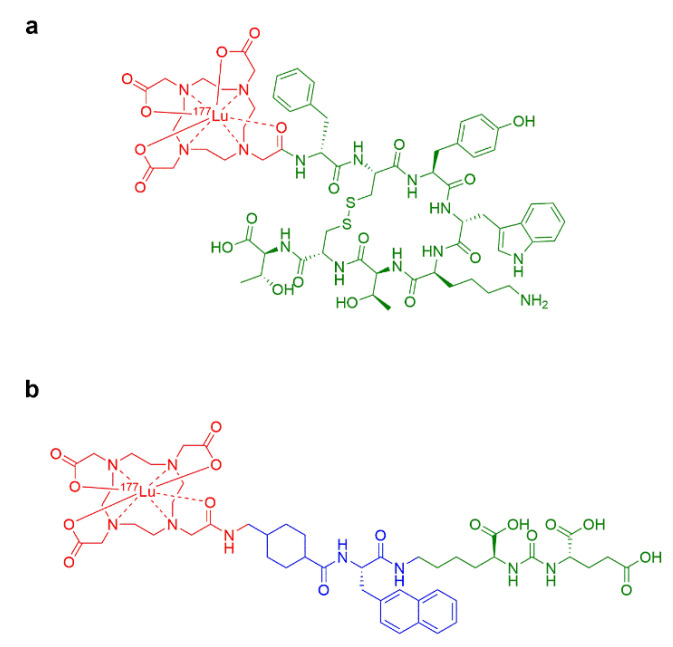
Chemical structures of the clinically approved Lutetium-177 based radiopharmaceuticals. (**a**) [^177^Lu]Lu-DOTATATE or Lutetium-177 oxodotreotide or Lutathera®. (**b**) [^177^Lu]Lu-PSMA-617 or Lutetium-177 vipivotide tetraxetan or Pluvicto®. Green, blue, and red colors indicate the ligand, linker, and ^177^Lu radiolabeled bifunctional chelator, respectively.

**Figure 2 pharmaceutics-15-01240-f002:**
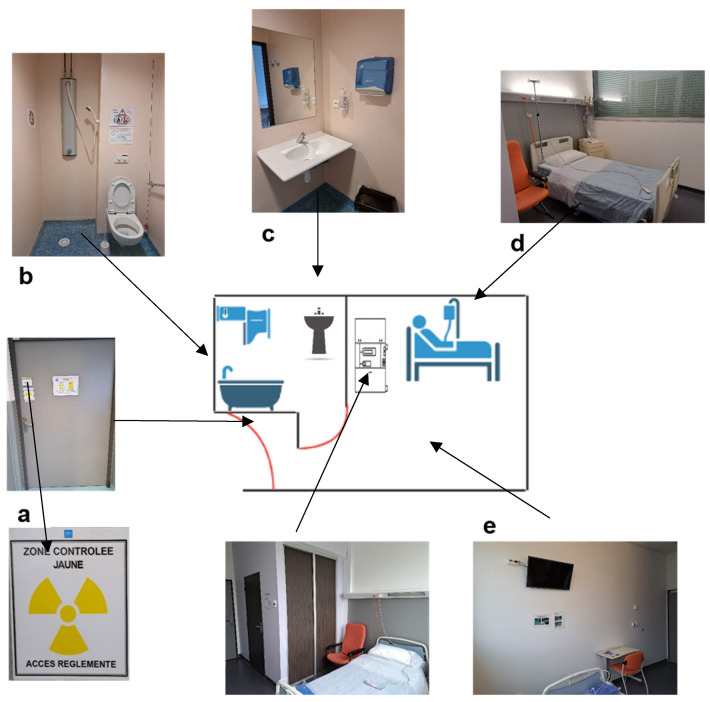
Scheme and photos of a radioprotected room in the therapy department of the Hospices Civils de Lyon, France. (**a**) Shielded door with zoning of the room (according to the different activities performed, here, the yellow controlled zone in the room was due to Iodine-131 therapy activity). (**b**) Shower with toilet connected to decay tanks for waste disposal. (**c**) Wash basin (connected to the decay tanks only due to Iodine-131 therapy activity). (**d**) Bed with single-use sheets. (**e**) Posting in the room and bathroom of the radiation protection rules for patients.

**Figure 3 pharmaceutics-15-01240-f003:**
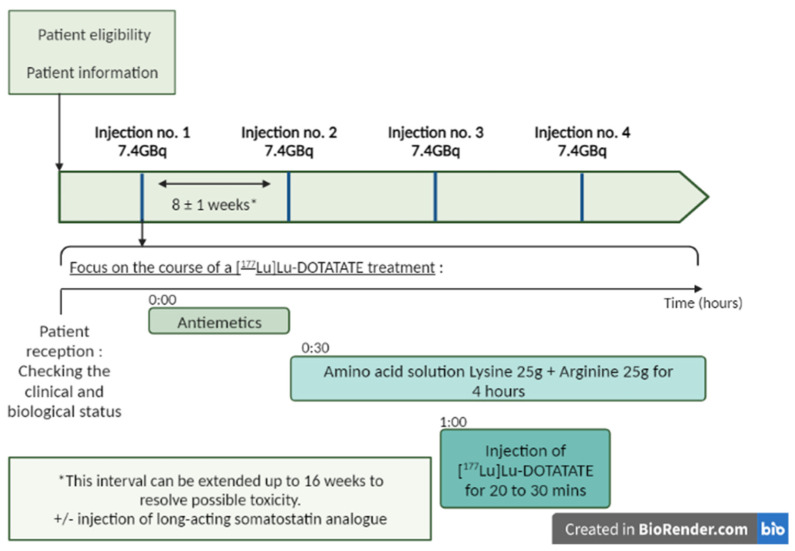
Treatment process for Lutathera®: four infusions of [^177^Lu]Lu-DOTATATE every 8 weeks (±1 week). Lutathera® administration should be preceded by an infusion of antiemetics and amino acids. The antiemetics were infused before the amino acids. The amino acids should be administered 30 min before [^177^Lu]Lu-DOTATATE and continue during treatment (see Section 2.3 for the details of premedication) For some patients, a long-acting somatostatin analog (Sandostatin LAR) may be administered intramuscularly (i.m.) every 3 to 4 weeks. A short-acting somatostatin analog may be used but should not be administered for 24 h prior to treatment with [^177^Lu]Lu-DOTATATE. Created in Biorender.com.

**Figure 4 pharmaceutics-15-01240-f004:**
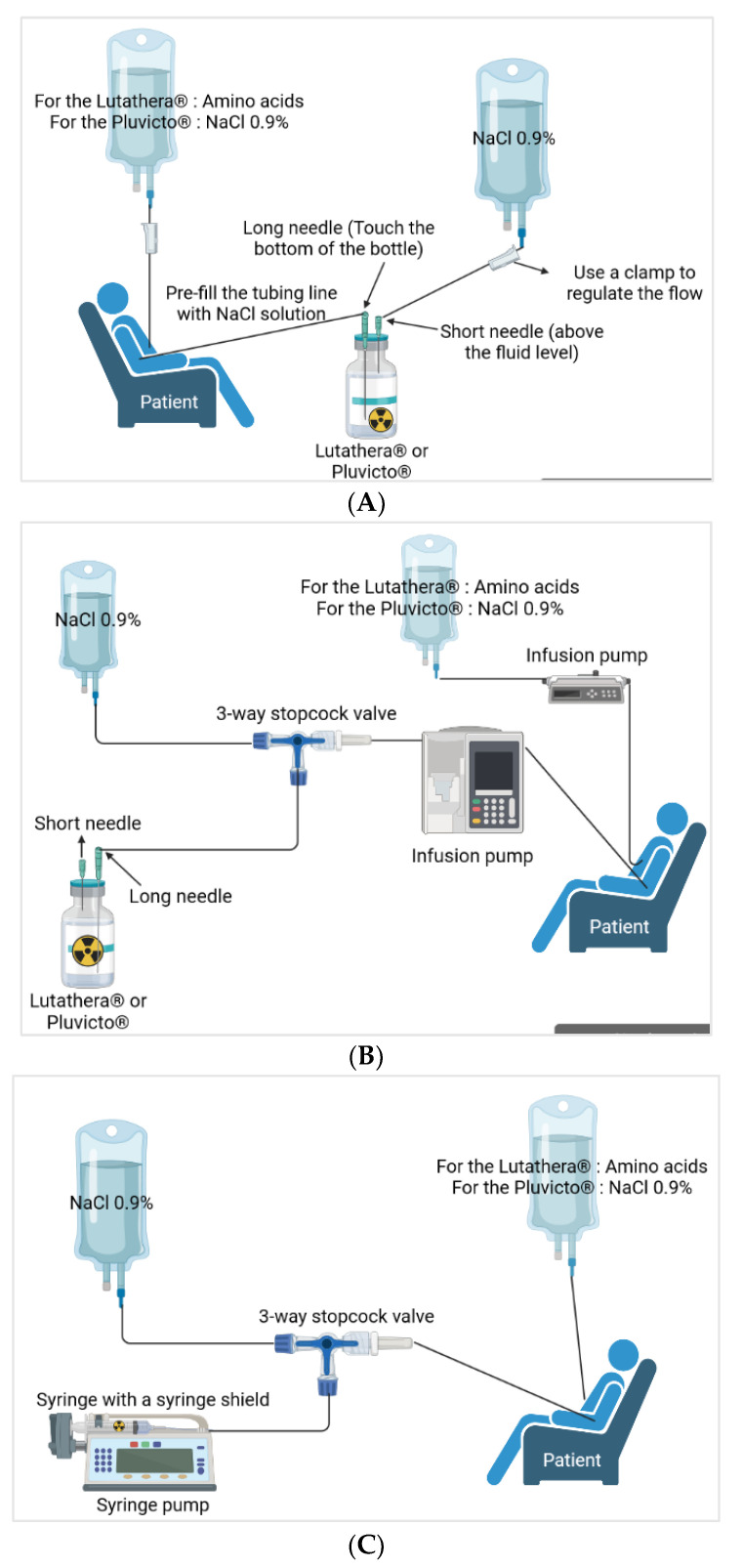
Administration procedure (modified from [59,61,63,65] and created in BioRender.com). (**A**) Gravity method, (**B**) pump method with vial, and (**C**) pump method with syringe.

**Figure 5 pharmaceutics-15-01240-f005:**
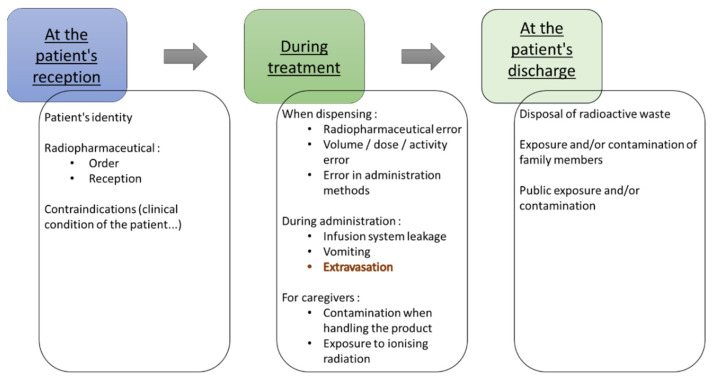
Examples of the risks that may occur during treatment with Lutathera® or Pluvicto®.

**Figure 6 pharmaceutics-15-01240-f006:**
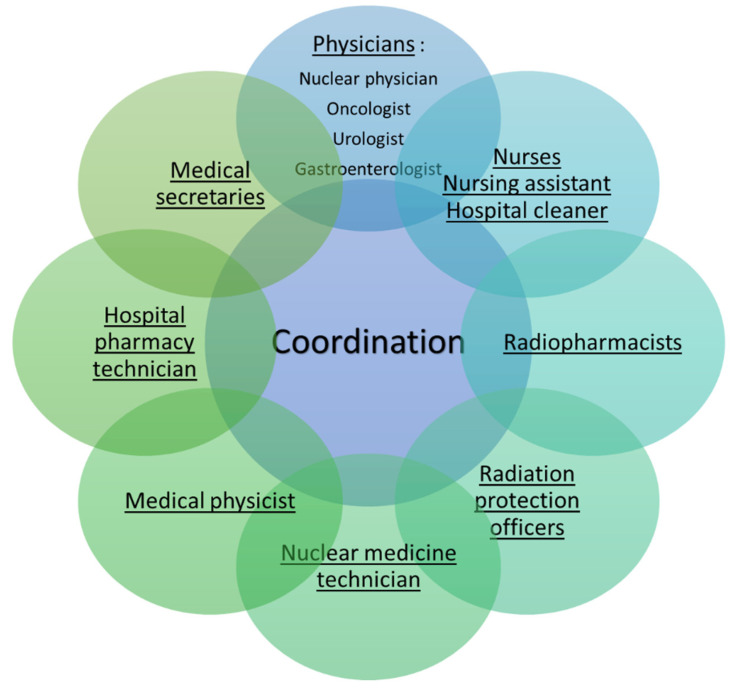
Coordination of all actors involved in the patient circuit for Lutathera® and Pluvicto®.

**Figure 7 pharmaceutics-15-01240-f007:**
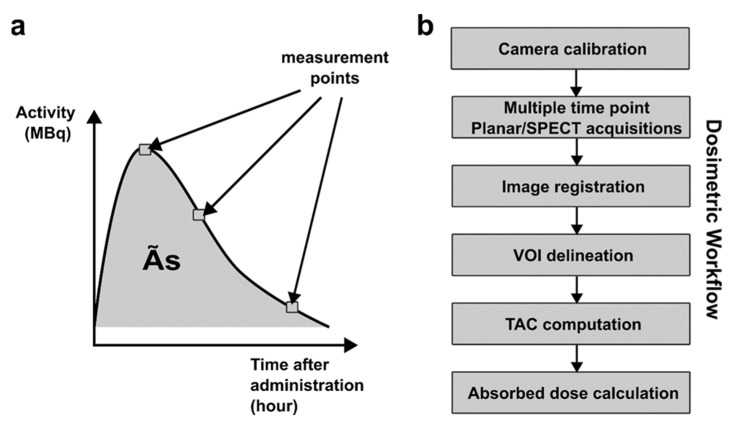
(**a**) Dosimetry methods for ^177^Lu radionuclide therapy: time activity curve determination with multi planar/SPECT measurements and (**b**) schematic representation of the clinical workflow.

**Table 1 pharmaceutics-15-01240-t001:** Dose modification schemes for Lutathera®.Adapted from [30].

Toxicity	Action	Result	Next Step
Grade 2 toxicity platelet count 40% increase in serum creatinine with a 40% decrease in clearance	Postpone treatment up to 16 weeks interval and monitor every 2 weeks	If resolution: re-treat with decrease of 50% radioactivity dose	No toxicity: full dose
		If persisting toxicity: stop treatment	Toxicity: stop treatment
Any Grade 3 (except pre-existing serum liver enzyme abnormalities)	Postpone treatment up to 16 weeks interval and monitor every 2 weeks	If resolution: re-treat with decrease of 50% radioactivity dose	No toxicity: full dose
		If persisting toxicity: stop treatment	Toxicity: stop treatment
Any Grade 4	Postpone treatment up to 16 weeks interval and monitor frequently. Act as necessary	If resolution: re-treat with decrease of 50% radioactivity dose	No toxicity: full dose
		If persisting toxicity: stop treatment	Toxicity: stop treatment

**Table 2 pharmaceutics-15-01240-t002:** Examples of dosage adjustments for Pluvicto® in the case of adverse events. Adapted from [48,59].

Adverse events	Severity	Action
Myelosuppression	Grade 2	Withhold Pluvicto® until improvement to Grade 1 or the baseline.
	Grade ≥3	Withhold Pluvicto® until improvement to Grade 1 or the baseline and reduce the Pluvicto® dose by 20%to 5.9 GBq (160 mCi).
	Recurrent Grade ≥3 after one dose reduction	Permanently discontinue Pluvicto®.
Renal toxicity	Confirmed serum creatinine increase (Grade ≥2) confirmed CLcr <30 mL/min	Withhold Pluvicto® until improvement.
	Confirmed ≥40% increase from baseline serum creatinine confirmed >40% decrease from baseline CLcr	Withhold Pluvicto® until improvement or a return to baseline. Reduce Pluvicto® dose by 20% to 5.9 GBq (160 mCi).
	Grade ≥3 renal toxicity and recurrent renal toxicity after one dose reduction	Permanently discontinue Pluvicto®.
Gastrointestinal toxicity	Grade ≥3	Withhold Pluvicto® until improvement to Grade 2 or the baseline. Reduce the Pluvicto® dose by 20% to 5.9 GBq (160 mCi).
	Recurrent Grade ≥3 gastrointestinal toxicity after one dose reduction	Permanently discontinue Pluvicto®.
Dry mouth	Grade 2	Withhold Pluvicto® until improvement or a return to the baseline. Consider reducing the Pluvicto® dose by 20% to 5.9 GBq (160 mCi).
	Grade 3	Withhold Pluvicto® until improvement or a return to the baseline. Reduce the Pluvicto® dose by 20% to 5.9 GBq (160 mCi).
	Recurrent Grade 3 dry mouth after one dose reduction	Permanently discontinue Pluvicto®.
Fatigue	Grade ≥3	Withhold Pluvicto® until improvement to Grade 2 or the baseline.

**Table 3 pharmaceutics-15-01240-t003:** The main radiation protection instructions for patients.

[^177^Lu]Lu-DOTATATE (Lutathera®)	[^177^Lu]Lu-vipivotide tetraxetan (Pluvicto®)
Drink plenty of water on the day of administration of the radiopharmaceutical and for 2 days afterwards to facilitate elimination.	
2. Urinate as often as possible after administration and defecate every day, using laxatives if necessary.	
3. Use the toilet in a sitting position and use toilet paper each time. If possible, flush the toilet twice.	
4. Wash your hands regularly to avoid contamination of surfaces.	
5. Avoid close contact with your family and try to keep a distance of at least 1 m for 7 days after administration.	5. Limit close contact to less than 1 m with your family for 2 days.
6. Limit close contact to less than 1 m with pregnant women and children for 7 days.	
7. Sleep in a separate bed from your partner for 7 days (or keep a distance of 2 m).	7. Sleep in a separate bed from your partner for 3 days (or keep a distance of 2 m).
8. Extend the period to 15 days if your partner is pregnant.	
9. Take a shower every day.	
10. Wash your clothes, underwear and sheets separately from other family members’ clothes using a standard wash cycle. If body fluids are present, use disposable gloves.	
11. Dispose of all items that have been soiled with body fluids either in the toilet if possible or in a separate garbage bag from other household waste.	
12. Abstain from sexual intercourse for 7 days after treatment.	
13. Use of effective contraception during treatment and for 6 months afterwards.	13. Use of effective contraception during treatment and for 14 weeks after the last dose.
14. Avoid travelling on public transport for the first few days after treatment if possible, otherwise maintain a distance of at least 1 m from the public.	
15. Keep your discharge letter with you for 3 months (hospitalization, travel...).	

## Data Availability

Not applicable.

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
