# Peer review of "Safety and Therapeutic Optimization of Lutetium-177 Based Radiopharmaceuticals"

_pharmaceutics, 2023, doi:10.3390/pharmaceutics15041240_

Round 1

Reviewer 1 Report

The manuscript "Safety and therapeutic optimization of Lutetium-177 based radiopharmaceuticals" is well organized and be read easily. The title and abstract cover the main aspect of the work. The introduction provides background and relevant information. The figures are clear and legible. The conclusion provides a clear summary of the main points.

In the main manuscript, there are some areas that can be improved:

Line 108, remove an n from bifunctionnal.

Complete the sentence on line 173, 175, 484

Reviewer 2 Report

The manuscript reviews the current status of Lutetium-177 radioisotope for the treatment of  cancer. The use of 177Lu is rapidly expanding worldwide, and a review is timely. The review is based on a large number of studies on the various aspects of 177Lu based cancer treatments. I found it very detailed and well organised. As my field is not nuclear medicine, I only can comment on parts of the manuscript on  nuclear decay and dosimetry.

I particularly appreciated section 5 on personalised dosimetry, which highlighted the importance to improve the estimation of the required dose for a particular patient. It could be an important goal, as it could reduce the required activity to be used and at the same time the risk of accidental radiation of people in contact with the patient. 

The nuclear decay data of 177Lu is given on page 2, line 64 to 77. I would recommended to use the most recent adopted values available at the US Nuclear Data Centre (https://www.nndc.bnl.gov/) or at the IAEA web site (https://www-nds.iaea.org/relnsd/vcharthtml/VChartHTML.html).  Specific values should be changed in the manuscript: T1/2=6.64 days, g-photon intensities: 208 keV 10.4%, 113 keV 6.2%. It might only have minimal impact on the dosimetry calculations of 177Lu.

Overall, I would be happy to recommend the paper for acceptance, but the corrections to the decay data of 177Lu is highly recommended.

Reviewer 3 Report

Manuscript Number: pharmaceutics-2253708

The manuscript “Safety and therapeutic optimization of Lutetium-177 based radiopharmaceuticals” by Typhanie Ladrière et al. describes various approaches to enhance the risk-benefit trade-off of radioligand therapy using clinically tested and reported tailored methods. The review focuses on using Peptide receptor radionuclide therapy using Lutetium-177-based radiopharmaceuticals in nuclear medicine and oncology, allowing personalized medicine. In addition, the review aims to assist clinicians and nuclear medicine staff in setting up safe and optimized procedures using approved 177Lu-based radiopharmaceuticals by highlighting the reported efficacy of 177Lu radiopharmaceuticals and the need for data on patient safety and management.

The manuscript is well written and organized and deserves publication in Pharmaceutics as a review.

Some minor points:

 Line 42-45: Since the historical application in the 1940’s of the theranostic radionuclides couple Iodine-123/Iodine-131 to diagnose and treat thyroid cancers, not other radioisotope based theranostics approaches (radiotheranostics) were approved for clinical use until recently [1].” I would strongly suggest the removal of the mention of iodine-123 since it was developed later in time and not initially part of the historical application of theranostic radionuclides in the 1940s. Instead, you could focus on the original Iodine-131 and its use in diagnosing and treating thyroid cancers. Furthermore, I would like to suggest the addition of other approved theranostic radionuclides, such as 99mTc and 186Re (99mTc-MDP, 99mTc-HDP, 99mTc-HEDP /186Re-HEDP), which have shown promise in clinical use. Including these newer options would provide a more comprehensive overview of the current state of theranostics.

Line 97: “…higher response rate that patients treated with...” change to “…than…”

Line 270 and 280: Table 2 is missing from the text and Tables.

Line 296: “There are different types of installation, the most frequently performed [45].” The sentence is incomplete.

Line 307-330: The text in images A, B and C is not so clear.

Line 392: Change Table 1 with Table 4.
